# Development of Glypican-3 Targeting Immunotoxins for the Treatment of Liver Cancer: An Update

**DOI:** 10.3390/biom10060934

**Published:** 2020-06-20

**Authors:** Bryan D. Fleming, Mitchell Ho

**Affiliations:** Laboratory of Molecular Biology, Center for Cancer Research, National Cancer Institute, National Institutes of Health, Bethesda, MD 20892-4264, USA; bryan.fleming@nih.gov

**Keywords:** recombinant immunotoxin, glypican-3, hepatocellular carcinoma, albumin binding domain, single-domain antibody

## Abstract

Hepatocellular carcinoma (HCC) accounts for most liver cancers and represents one of the deadliest cancers in the world. Despite the global demand for liver cancer treatments, there remain few options available. The U.S. Food and Drug Administration (FDA) recently approved Lumoxiti, a CD22-targeting immunotoxin, as a treatment for patients with hairy cell leukemia. This approval helps to demonstrate the potential role that immunotoxins can play in the cancer therapeutics pipeline. However, concerns have been raised about the use of immunotoxins, including their high immunogenicity and short half-life, in particular for treating solid tumors such as liver cancer. This review provides an overview of recent efforts to develop a glypican-3 (GPC3) targeting immunotoxin for treating HCC, including strategies to deimmunize immunotoxins by removing B- or T-cell epitopes on the bacterial toxin and to improve the serum half-life of immunotoxins by incorporating an albumin binding domain.

## 1. Introduction

Liver cancer remains one of the deadliest cancers in the world despite recent advances in anti-cancer therapeutics [1]. Roughly 780,000 deaths, or about 8% of worldwide cancer-related deaths, are attributed to liver cancer [2]. There were over 840,000 new cases of liver cancer reported in 2018, with the majority of these new cases occurring in Asia [2]. Males are disproportionately affected by liver cancer, with cases diagnosed about three times more often than females [2]. Risk factors include heavy alcohol intake, smoking, obesity, type 2 diabetes and chronic hepatitis B or C infection [2,3,4,5]. In the United States, there are about 42,000 new cases a year, with the incidence of liver cancer growing faster than any other cancer type [2,6,7,8]. Compared to other cancer types, liver cancer patients diagnosed in the United States are met with one of the lowest five-year survival averages (30%) [7]. Diagnosis of liver cancer typically requires imaging with a computerized tomography (CT) or magnetic resonance imaging (MRI) scan, making it difficult to properly diagnose patients in rural settings [4]. If liver cancer is diagnosed in the early stages, surgical removal or liver transplantation represent the treatment options with the best overall chance of survival [4,9]. Unfortunately, liver cancer often goes undetected until the late stages of cancer, which limits treatment options [1,4]. The U.S. Food and Drug Administration (FDA) has approved the tyrosine kinase inhibitors sorafenib and regorafenib as first- and second-line therapies, respectively [4,10,11,12,13,14]. However, these drugs only provide about a three-month increase in overall survival [10,13]. The development of new therapies to treat liver cancer are desperately needed.

Hepatocellular carcinoma (HCC) is the most common form of primary liver cancer and is responsible for 75–85% of all reported cases [2]. Genetic analysis of HCC revealed that the majority of cases are associated with an upregulation of glypican-3 (GPC3) [15,16,17,18]. GPC3 is a cell surface protein consisting of 580 amino acids with a molecular weight of around 70 kDa [18,19,20,21]. The C-terminus is decorated with two heparan sulfate chains and is attached to the cell surface by a glycosylphosphatidylinositol (GPI) anchor [18,20,21]. We and others have demonstrated that GPC3 is a suitable target for antibody-based therapeutics for several reasons [18]. First, as an oncofetal proteoglycan, GPC3 is primarily expressed in embryonic tissues and has virtually no expression in healthy human tissues [15,16,17,22,23,24,25,26,27,28]. Consequently, GPC3 has a high tumor specificity due to its upregulation in hepatocellular carcinoma and lack of expression in healthy human tissues. Second, GPC3 not only has a high surface expression level, but also has a high internalization rate [29,30]. These two factors help to ensure that sufficient immunotoxin enters the tumor cell. Finally, GPC3 has been shown to interact with several cell proliferation pathways, including the Wnt/β-catenin, Hedgehog, Yap, HGF and insulin-like growth factor signaling pathways [21,25,29,31,32,33,34,35,36,37,38,39]. Blocking the GPC3 interaction with Wnt has been shown to cause a down-regulation of cell proliferation in HCC cells [29,39]. Taken together, these three factors make GPC3 an intriguing target for HCC treatment.

The development of antibody-based therapeutics has been a growing trend over the last few decades [40,41,42]. Using an antibody domain to direct toxic drugs or immune cells can significantly decrease the occurrence of off-target side effects [40,41,43,44]. Treatment of leukemias have shown some of the most promising results due to well-defined surface cell lineage markers [45,46,47,48,49,50]. The FDA recently approved Lumoxiti, a CD22-targeting recombinant immunotoxin using *Pseudomonas* exotoxin A, as a treatment for patients with relapsed or refractory hairy cell leukemia that have received at least two prior systemic therapies [47,51]. This recent FDA approval helps to demonstrate the potential role recombinant immunotoxins can play in the cancer therapeutics pipeline. Antibody–drug conjugates are another class of drugs that have seen a recent surge in FDA approvals [43,44,52]. CAR-T cell based clinical trials have also shown a rapid increase recently, but this technology still has limited success in advanced solid tumors, carries a relatively high cost, and requires an extensive hospital stay [53,54,55,56]. Protein-based antibody drugs such as immunotoxins represent an off-the-shelf therapy that can be used to treat cancer patients in any hospital or clinic around the world. The unique mechanism of action for inhibiting protein synthesis in cancer cells makes immunotoxins a viable class of antibody drugs that are distinct from almost any other common antibody drug format, including bispecific antibodies and antibody–drug conjugates. These chimeric therapeutics combine an antibody’s antigen binding domain with the ribosomal inhibitory domain of the *Pseudomonas* exotoxin (PE) or other toxin domains, such as ricin and diphtheria toxin [57,58,59].

Domain III of the *Pseudomonas* exotoxin (PE) has been well established as the enzymatic domain of the exotoxin [60,61,62,63]. The domain functions by enzymatically modifying elongation factor 2 (EF2) through an ADP ribosylation reaction [64]. The inactivation of ribosomal function leads to the induction of cell death through the apoptotic pathway [63,64]. Frequently, domain II and domain III from the *Pseudomonas* exotoxin are used in immunotoxin construction and are often referred to as PE38 due to their molecular weight [29,65,66,67,68,69,70]. An immunotoxin’s ability to reduce cancer cell proliferation through the inhibition of protein synthesis allows for the development of potent anti-cancer therapeutics.

The use of immunotoxins in clinical trials has been rapidly growing and is revealing important insights into how these molecules behave in patients. Table 1 highlights some of various immunotoxin clinical trials listed as active or completed on the clinicaltrials.gov website. Most immunotoxins being tested in clinical trials use the PE-derived ribosomal inhibitory domain. The side effects associated with these immunotoxins most often present as mild vascular leak syndrome that can be controlled with corticosteroids [47,71,72]. In some PE-based immunotoxin clinical trials, there have been adverse effects reported following treatment. Cardiac arrythmias, pneumonitis, hemolytic uremic syndrome, elevated liver enzymes and sepsis are some of the more severe side effects associated with immunotoxin treatment [73,74,75]. Other ribosomal inhibitor domains are also being explored in these current clinical trials. Diphtheria toxin and ricin derived inhibitory domains are the next two most frequently used. Diphtheria-derived toxins utilize the same elongation factor 2 modification for ribosomal inactivation as the PE-derived toxins [72,76]. Ricin toxin uses a different inhibition mechanism by cleaving the N-glycosidic bond of an adenine residue in the 28S rRNA [77]. Ricin toxins are known for their ability to be rapidly cleared from blood circulation, so most therapeutics now use a deglycosylated version of the A chain to help prevent toxin uptake by the liver [78]. This also helps to reduce some of the toxicities seen in the early clinical trials involving ricin [72,79]. Gelonin is another plant toxin that is being developed for immunotoxin use because it shares the same mechanism of action as ricin [80]. One common observance in these clinical trials is the formation of anti-toxin antibodies that is caused by the immunogenic nature of the toxins utilized [47,81,82]. Anti-toxin antibodies can reduce the effectiveness of the treatments by shortening the serum half-life and limiting the number of treatment cycles [47,71]. Therefore, we believe that decreasing the immunogenicity and increasing the serum half-life of immunotoxins will have the greatest impact on therapeutic efficacy.

## 2. Optimizing Antibody Targeting of GPC3

Our lab has generated three unique antibodies capable of targeting GPC3. These antibodies were generated by hybridoma [24] or phage display technology [25,93] and ultimately target different portions of the GPC3 molecule. An overview of the antibodies that our lab has generated, and their binding sites, can be found in Figure 1A. The first antibody we described was a mouse monoclonal antibody named YP7 [24]. A 50 amino acid fragment from the C-terminus (residues 511–560) was used to immunize BALB/c mice [24]. We now know the actual epitope for YP7 to be located between residues 521–530 [28]. This antibody was found to have an affinity of 0.3 nM on cells overexpressing GPC3 [24]. The next two antibodies our lab generated were identified using phage display technology. The HS20 antibody is a human monoclonal antibody that binds the heparan sulfate chains attached to serine 494 and serine 508 of GPC3 [94]. These heparan sulfate modifications are not limited to GPC3, so HS20′s therapeutic potential needs to be validated due to its potential off-tumor toxicities [95,96]. Interestingly, the HS20 antibody down regulates Wnt/β-catenin signaling by blocking Wnt binding to the heparan sulfate chains [37,95]. Like the HS20 antibody, HN3 was identified by phage display technology and can block Yap and Wnt signaling in HCC cells [25,29,39]. However, HN3 is different than HS20 because is a single domain antibody, also known as a nanobody. HN3 binds directly to the N-lobe of GPC3 and was recently shown to rely heavily on an interaction with residue F41 of GPC3 [39]. HN3 showed a high affinity of 0.7 nM, which is about half the measured affinity of YP7 [25]. Both of these antibodies were used to generate PE38-based immunotoxins and their activity was tested in mouse xenograft models [29]. Despite the higher affinity associated with the YP7 antibody, it was HN3-PE38 that performed the best in these studies [29]. This was attributed to the synergistic effect of blocking both Wnt signaling and the inhibition of protein synthesis [29]. Due to the high activity we observed with HN3-PE38, we decided to further explore this immunotoxin for clinical development.

The use of nanobodies provides a unique platform for immunotoxin design. The small size of nanobodies allows them to potentially seek out buried epitopes by binding inside of protein clefts that may be unreachable by conventional antibodies [39,42,93,97]. Our protein modeling revealed the presence of two hydrophobic grooves located on the N-lobe of GPC3 for potential Wnt binding [39]. We did a number of point mutations in these two predicted regions and identified one of the regions was the actual Wnt binding domain. Interestingly, mutation of the phenylalanine (F) at position 41 to a hydrophilic glutamic acid (E) in this region was found to disrupt the binding of Wnt3a and reduce Wnt activation in a functional β-catenin signaling reporter assay [39]. We also found that this same F41E mutation reduced HN3 binding for GPC3, supporting the fact that HN3 directly competes with Wnt for GPC3 binding [39].

In addition to HN3-derived immunotoxins, our group previously generated the SD1 nanobody-based immunotoxin (SD1-PE38) and showed that it inhibited the proliferation of mesothelin-positive mesothelioma (NCI-H226) and cholangiocarcinoma (KMBC) cell lines [98]. Several other labs are also exploring the use of nanobodies for the development of recombinant immunotoxins. One group has reported positive results using a tandem, humanized, anti-CD7 camel V_H_H nanobody fused to PE to treat T cell acute lymphoblastic leukemia in a mouse xenograft model [99,100].

Another advantage of using nanobodies for the construction of immunotoxins is their high level of solubility and ease of refolding. One group has reported the use of transplastomic lettuce as a bioreactor to produce biologically active immunotoxin containing a camel V_H_H domain targeting VEGFR2 fused to PE [101,102]. We find that HN3 displays good solubility and has a high refoldability. We generally see HN3 immunotoxin protein recovery of 10–20% after column purification, while other immunotoxins can have recoveries as low as 2% [103,104]. These two advantages are another reason why the HN3 antibody was the best choice for therapeutic development.

Another aspect of HN3 binding we explored was to determine the specificity of our nanobody. Side effects can occur if the target antigen is expressed on healthy human tissue. In a clinical trial using anti-mesothelin immunotoxins, the majority of side effects were attributed to on-target, off-cancer cell binding due to mesothelin’s expression on healthy tissue [81,105]. To help anticipate any off-target side effects of HN3, we utilized a cryopreserved human tissue microarray that retained the conformational epitope needed for HN3 binding. HN3 staining showed little to no binding on healthy adult liver tissues [27]. Importantly, vital organs such as the heart, brain, lungs and kidney were also found to be negative for HN3 staining [27]. These results suggest that HN3-based immunotoxins will show low off-target binding in patients. It also supports previous reports that GPC3 serves as a strong diagnostic marker for HCC development [16,18].

We also tested the ability of our HN3-based immunotoxin to display cross-species binding to mouse GPC3. HN3 immunotoxin displayed a high affinity for both human (4.9 nM) and mouse (5.8 nM) GPC3 when tested with an Octet system [27]. This suggests that the Wnt binding domain we have identified on the N-lobe of GPC3 is highly conserved between both human and mouse GPC3. Interestingly, there are currently contrasting reports about the role of GPC3 in human and mice. The overexpression of mouse GPC3 has been associated with a reduction in liver proliferation and HCC formation, indicating growth and regulatory factors other than mouse GPC3 are required for HCC formation [106]. However, our research has shown that when human derived Hep3B are subject to GPC3 knockdown or knockout, they grow much more slowly on culture dishes and fail to form tumors in mouse xenograft models [25,39]. These results highlight the importance of GPC3 upregulation in the formation of hepatocellular carcinomas and support the idea of GPC3 as a strong therapeutic target for hepatocellular carcinoma.

## 3. Reducing the Immunogenicity of the Toxin Domain

Addressing the immunogenicity of the PE has been one of the biggest challenges in preparing immunotoxins for the clinical environment. The first attempt at generating a deimmunized immunotoxin involved the identification and mutation of B cell epitopes [103,107,108]. Mice were immunized with various PE domains and a panel of 60 antibodies with PE38 reactivity were isolated [103]. Exotoxin domains containing a single point mutation were then used to create a topographical epitope map [103]. A total of seven major epitope regions were identified that spanned domain II and domain III [103]. Several of these epitopes were deleted when domain II was replaced with a furin-cleavable linker [109]. The removal of domain II deleted a large number of protease sites, resulting in an immunotoxin that was less susceptible to lysosomal degradation [109]. This study showed that domain II was not essential for immunotoxin function, although some cell types showed higher sensitivities to immunotoxins containing domain II [109]. The other B cell epitopes were silenced by combining the point mutations found to have the greatest impact on antibody binding [103,108,110]. A similar strategy to reduce B cell antigenicity in diphtheria toxin through the selective mutation of arginine, lysine, aspartate, glutamate and glutamine residues has been reported [111]. This deimmunized anti-CD22/CD19 diphtheria toxin-based immunotoxin was found to induce lower antibody responses in BALB/c mice while retaining the ability to inhibition B cell malignancies in vitro [111].

We incorporated the B cell deimmunized toxin and furin-cleavable linker into our HN3 immunotoxin to create HN3-mPE24 (Figure 1B) [112]. An overview of the exotoxin domains we used in this paper can be found in Figure 1B. An analysis of enzymatic activity revealed that 89% of the wild-type enzymatic activity was retained after the mutations [27]. When HN3-mPE24 was tested in a xenograft model using nude mice, it was found to be much less toxic than the original HN3-PE38 version [27,112]. The ability to give HN3-mPE24 at doses up to 10 mg/kg resulted in better tumor regression than was possible with the HN3-PE38 immunotoxin, which had a maximum tolerated dose of 0.8 mg/kg [27,29]. The generation of HN3-mPE24 provided a proof-of-concept study that HN3 could be successfully combined with a furin-cleavable linker and a deimmunized domain III to generate an immunotoxin [112].

Another strategy for the deimmunization of immunotoxins focused on reducing the T cell response through the elimination of CD4+ T cell activating antigenic peptides [113,114,115]. The hypothesis was that a reduction in helper T cells activation would result in a weaker overall immune response to the exotoxin. Peripheral blood mononuclear cells (PBMCs) from 50 healthy donors were stimulated with 111 overlapping peptide sequences spanning the entirety of the PE38 toxin domain, then their activation was determined by ELIspot assay [113]. A total of eight major epitopes that caused strong PBMC activation were identified in this screen [113]. A similar point mutation-based scan was used to identify the key amino acid residues responsible for T cell activation [113]. The six epitopes found in domain III were mutated to create the T cell deimmunized T20 domain (Figure 1B) [114]. This variant was shown to reduce neutralizing and anti-drug antibodies in mice [116]. Similar to mPE24, T20 retained a high level of cytotoxic activity when combined with HN3 in deimmunized immunotoxins (HN3-T20 and HN3-mPE24) [27]. ADP ribosylation activity testing of HN3-T20 revealed that there was a 27% reduction in activity compared to PE24, the wild-type domain III of PE. This was greater than the reduction we observed with HN3-mPE24 (11%), despite having one fewer point mutation [27].

In addition to the single deimmunization route, combining the B cell and T cell silencing mutations into a single exotoxin domain has been explored [27,117]. We generated HN3-T19 and HN3-M11 which contained a total of 10 and 11 point mutations of B and T cell epitopes, respectively (Figure 1B) [27]. Overall, these two immunotoxins showed a significant reduction in ADP ribosylation function with HN3-T19 retaining 55% activity and HN3-M11 only retaining 11% activity [27]. When we used our deimmunized HN3 immunotoxins (HN3-mPE24, HN3-T20, and HN3-T19) to treat Hep3B xenografts in mice, we discovered that HN3-T20 caused the greatest increase in overall survival [27]. We found this surprising due to the fact that HN3-mPE24 had shown the highest level of activity in our in vitro screens [27]. Due to the better performance in the animal model, we suggested that HN3-T20 represented a better candidate for clinical development.

## 4. Increasing the Serum Half-Life of Immunotoxins

The short half-life of recombinant immunotoxins is perhaps the second biggest challenge for immunotoxins in the clinical environment. Pharmacokinetic studies on our HN3-T20 immunotoxin revealed a half-life of around seven minutes in nude mice [27]. It has been reported immunotoxin half-lives range from 8 to 13 minutes [118]. This is much shorter than the 44-hours half-life that has been reported for trivalent nanobodies that share a similar size [119]. Our HN3-based PE24 immunotoxins have a relatively small size of about 39 kDa [27]. We believe that this small size can help increase tissue permeability, but this may also have a negative impact on overall serum retention. Research has shown that the addition of an albumin binding domain has the ability to improve the overall serum retention of immunotoxins [27,118,120].

We tested an albumin binding domain derived from the *Streptococcal* protein G (ABD) and a second one isolated from a llama V_H_H antibody (ALB1) to identify which would provide the greatest benefit for our immunotoxin [27]. The addition of the albumin binding domains had little effect on HN3 binding or overall in vitro activity [27]. However, both of these albumin binding immunotoxins showed significant improvement in in vivo activity [27]. This increased in vivo efficacy was attributed to an increase in serum half-life. HN3-ABD-T20 displayed a 44-fold increase in serum retention and was found to have a 326-minutes half-life in nude mice [27]. HN3-ALB1-T20 was found to have a half-life of 164 minutes, representing only a 22-fold improvement in nude mice [27]. It is important to note that the increase in serum half-life also appears to be associated with increased off-target toxicities. The maximum tolerated doses we observed for the albumin binding immunotoxins were significantly less than the original HN3-T20 immunotoxin [27].

Our observed improvement of serum half-life was consistent with other reports that showed about a 10-fold increase in half-life with the addition of an albumin binding domain [118]. Our increases in serum half-life corresponded well with the observed improvements in our in vivo model, since the ABD domain showed about a two-fold higher affinity for mouse serum albumin than the ALB1 domain [27]. It is possible that HN3-ALB1-T20 might perform better in patients because it showed the highest overall affinity for human serum albumin [27]. Using transgenic mice that express human serum albumin for future studies may provide valuable insights into how our immunotoxins will behave in patients [121]. Additionally, the bacterial origin of the ABD domain may illicit an immune response like those associated with the toxin domain. The llama-derived ALB1 has the potential to be less immunogenic due to the fact that camelid antibodies share a high similarity to human antibody frameworks [122].

Another method for increased serum half-life involves site-directed PEGylation of the immunotoxin [123]. This increase in serum half-life was shown to cause a substantial improvement in anti-tumor activity [123]. As mentioned before, the foreign nature of the albumin binding domains may trigger unwanted immune responses. Some reports have suggested that PEGylation is associated with a decrease in protein immunogenicity [124,125]. However, increased IgM production following repeated doses of PEGylated liposomes results in accelerated blood clearance with subsequent doses [126]. Additionally, anti-PEG antibodies can be detected in about 25% of healthy patients, most likely due to environmental exposure to PEG-containing compounds used in the cosmetics, pharmaceutical and food-processing industries [127]. Whether this strategy will benefit our HN3-based immunotoxins needs to be tested.

## 5. Perspective and Future Direction

This review discussed the recent efforts in developing an anti-glypican-3 immunotoxin for clinical use. We have recently shown that the HN3 nanobody displays low off-target staining in complex human tissues and that it directly competes for the Wnt binding site on GPC3. This direct inhibition of Wnt binding leads to a decrease in β-catenin signaling and reduced liver cancer proliferation. To help optimize the toxin domain, we combined HN3 with six different variants of the *Pseudomonas* exotoxin. We tested the effects of deimmunization strategies and a domain II deletion on enzymatic activity and cancer cell growth inhibition. The HN3-mPE24 and HN3-T20 deimmunized immunotoxins performed the best in our preclinical screen, retaining much of their function. HN3-T20 had the best overall results in vivo, causing the greatest increase in overall survival. Further improvements to the HN3-T20 immunotoxin were tested to improve serum half-life. We compared an ABD domain isolated from *Streptococcal* protein G and the ALB1 domain isolated from a llama nanobody in our HN3 immunotoxins. The bacterial-derived ABD resulted in greatly improved serum half-life, with low levels of circulating immunotoxin being detected after a 24-hour incubation. With these recent improvements in immunotoxin function, we believe that HN3-based immunotoxins represent an alternative therapeutic option for patients that have not responded to traditional therapies.

Clinical trials will provide valuable information about how our deimmunized immunotoxins behave in patients. The point mutations introduced to silence antigenic epitopes are most likely MHC-specific mutations. The large variations in global MHC molecules means that a single deimmunized molecule may work for some individuals and not for others. These same point mutations might in fact increase antigenicity when the MHC background is changed. The complexity of the human immune response most likely means that a universally deimmunized toxin domain is extremely challenging. Gaining a better understanding of how patients from diverse immunological backgrounds respond to immunotoxin therapy would provide important insights. Combining drugs that are involved in immune modulation with immunotoxin treatment may also play a role in the reduction of the immunogenicity of immunotoxins in the clinical setting [71,128].

Nanobodies represents an ideal platform for the production of bispecific or even trispecific therapeutics [129,130,131]. The creation of HN3-ALB1-T20 is a good example of how two nanobody domains allow for the easy creation of bispecific immunotoxins [27]. The incorporation of additional albumin binding domains or cancer-binding nanobodies could be explored to further improve HN3 immunotoxin function. Other labs have reported the construction of bispecific immunotoxins for targeting B cell malignancies by targeting CD19 and CD22 simultaneously [91]. Further optimization of our HN3 immunotoxins has the potential to increase efficacy, expand the cancer targets and help provide a new cost-effective therapeutic for liver cancer patients.

It is important to note that the use of our HN3-based immunotoxins may not be limited to the treatment of hepatocellular carcinomas. While the upregulation of glypican-3 has been well established in HCC samples, there is growing evidence that GPC3 may in fact be upregulated in a diverse array of cancers. It has been reported that about half of squamous cell lung cancers [132,133,134] and about 20% of head and neck squamous cell cancers [133,135] are associated with increased GPC3 levels. Additionally, expression of GPC3 has been detected at low frequency in breast [136,137,138], thyroid [139], gastric [140], ovarian [141,142], colorectal [135] and some pediatric [143] cancers. The exact role of GPC3 in these cancers is still being investigated. There have been some reports that GPC3 expression in breast cancer is associated with a reduction in cancer invasion and cell metastasis [137,138]. Whether treatment with anti-GPC3 immunotoxins will be beneficial in these cancer types is still unknown. It would be interesting to screen additional cancers in the future to identify those that may be sensitive to HN3 immunotoxin therapy.

## Figures and Tables

**Figure 1 biomolecules-10-00934-f001:**
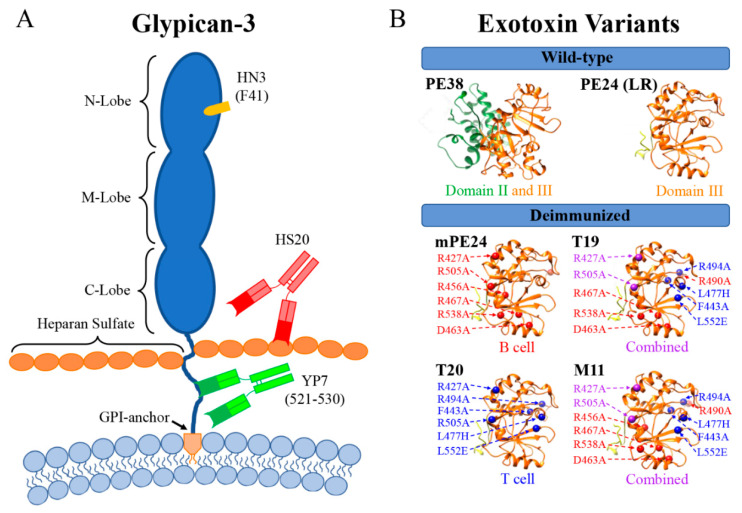
(**A**) Illustration of glypican-3 and our various anti-glypican-3 antibodies. The approximate binding epitope of the key residue for the antibodies are indicated in the parentheses. (**B**) Schematic of the various *Pseudomonas* exotoxin domains discussed in this review. Exotoxins labeled as combined have both B cell and T cell deimmunizations. The mutations indicated in purple were common to both B cell and T cell deimmunizations.

**Table 1 biomolecules-10-00934-t001:** Summary of the current clinical trials exploring the use of immunotoxins.

NCT Number	Drug Name	Target	Toxin Source	Phase	Cancer Type/Disease	Status	References
01829711	Moxetumomab pasudotox	CD22	PE	III	HCL	Comp	Kreitman et al. 2018 [51]
00074048	BL22	CD22	PE	II	Leukemia	Comp	Kreitman et al. 2009 [83]
00001805	LMB-1 + rituximab	Lewis Y	PE	II	[*a*]	Comp	Hassan et al. 2004 [84]
00924170	LMB-2 + chemotherapy	CD25 (anti-Tac)	PE	I/II	ATL	Active	Kreitman et al. 2015 [85]
02810418	LMB-100 +nab-Paclitaxel	Mesothelin	PE	I/II	Pancreatic	Active	Alewine et al. 2019 [73]
00006268	Cintredekin besudotox	IL-13 receptor	PE	I/II	Brain, CNS	Comp	Kunwar et al. 2007 [86]
02219893	MOC31PE	EpCAM	PE	I/II	Colorectal	Comp	Frøysnes et al. 2017 [87]
01445392	SS1(dsFv)-PE38 + chemo	Mesothelin	PE	I/II	Mesothelioma	Comp	Hassan et al. 2014 [88]
00104091	TP38	EGFR	PE	II	Glioblastoma	Comp	Sampson et al. 2008 [89]
00611208	A-dmDT390-bisFv(UCHT1)	CD3	DT	II	Leukemia, Lymphoma	Comp	Frankel et al. 2015 [90]
00211185	Denileukin Diftitox	CD25 (anti-Tac)	DT	II	Lymphoma	Comp	Foss et al. 2013 [75]
00889408	DT2219ARL	CD19/CD22	DT	I/II	Lymphoma, Leukemia	Comp	Bachanova et al. 2015 [91]
02027805	T-Guard	CD3/CD7	Ricin	I/II	GVHD	Comp	Groth et al. 2019 [92]
00038051	Hum-195/rGel	CD33	Gelonin	I	[*b*]	Comp	Borthakur et al. 2012 [80]

ATL, Adult T-cell Leukemia; CNS, Central Nervous System; Comp, Complete; DT, Diphtheria toxin; GVHD, Graft vs. Host Disease; HCL, Hairy Cell Leukemia; PE, *Pseudomonas* exotoxin. [*a*] Breast, Colon, Lung, Pancreatic, Stomach; [*b*] Acute Myeloid Leukemia, Chronic Myelomonocytic Leukemia.

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
