# Peer review of "Development of Glypican-3 Targeting Immunotoxins for the Treatment of Liver Cancer: An Update"

_biomolecules, 2020, doi:10.3390/biom10060934_

Round 1
Reviewer 1 Report
Development of Glypican-3 Targeting Immunotoxins for the Treatment of Liver Cancer: An Update
In this review, the authors describe recent advances in the development of anti-GPC3 recombinant immunotoxins for the treatment of hepatocellular carcinoma, updating a review published in 2016 by the same authors. The focus is mainly on the deimmunization and half-life extension of anti-GPC3 RITs based on Pseudomonas exotoxin A. In general, the quality of the article is good, and it does provide updated information on anti-GPC3 RITs, but there have not been any major advances since the previous review. Out of 130 citations, 38 were published in 2017 or later (after the 2016 review article). Perhaps 6 of these articles are primary research related directly to the development of anti-GPC3 RITs. The lack of major advances and the short window of time since the last review limits the potential impact of the article. Overall, the article is acceptable for publication with revisions, but its contribution to the field is limited.
A list of some additional comments is below. I have made editorial suggestions to improve clarity and readability on the paper itself, included as an addendum to my formal comments here.
Major:
A schematic figure representing the different PE fragments and RITs that the authors discuss is needed in the interest of clarity.
The Perspective and Future Directions section seems lacking. Perhaps a summary of the immunogencity/half-life sections and some thoughts about bringing the anti-GPC3 RITs into the clinic would help to fill it out. It would also be helpful to emphasize and expand on what has changed since the last review.
Minor:
Table 1: formatting is problematic. Please revise so that words are not broken across multiple lines.
Line 87: Start a new paragraph
Lines 124-126: It is not clear to me how an antibody against heparan sulfate would be specific for GPC3. Wouldn’t it react with any HS? Can you add some details to clarify the specificity of HS20?
Line 150: Start a new paragraph
Line 156: Start a new paragraph
Line 173: Start a new paragraph
Lines 196-197: It is not entirely true that the removal of domain II did not affect RIT function. It depends on several factors such as cell type and target antigen. More accurately, one could say that domain II was not essential to immunotoxin function or something similar.
Line 202: Start a new paragraph
Line 223: Don’t change the nomenclature here by referring to “LR”. Keep previous PE24 nomenclature used elsewhere in this manuscript on lines 205/207/209/220/222/224/230/232.
Line 225: Start a new paragraph
Lines 249 & 250: please specify the origins of the different albumin-binding domains to clarify the abbreviations.
Lines 250 & 251: please specify the model system in which the half-life study was conducted. Nude mice?
Lines 271-272: These statements belong in the immunogenicity section.
Lines 273-279: This paragraph belongs in the future directions section.
Lines 281-282: This sentence does not fit with the remainder of the paragraph.
Lines 282-288: This information seems out of context and would be more appropriate after a brief summary of the immunogenicity section.
Citations: #39 & #40 appear to be identical. Please confirm that they are separate references.

Author Response
Reviewer’s 1 Comments:
Development of Glypican-3 Targeting Immunotoxins for the Treatment of Liver Cancer: An Update
In this review, the authors describe recent advances in the development of anti-GPC3 recombinant immunotoxins for the treatment of hepatocellular carcinoma, updating a review published in 2016 by the same authors. The focus is mainly on the deimmunization and half-life extension of anti-GPC3 RITs based on Pseudomonas exotoxin A. In general, the quality of the article is good, and it does provide updated information on anti-GPC3 RITs, but there have not been any major advances since the previous review. Out of 130 citations, 38 were published in 2017 or later (after the 2016 review article). Perhaps 6 of these articles are primary research related directly to the development of anti-GPC3 RITs. The lack of major advances and the short window of time since the last review limits the potential impact of the article. Overall, the article is acceptable for publication with revisions, but its contribution to the field is limited.
A list of some additional comments is below. I have made editorial suggestions to improve clarity and readability on the paper itself, included as an addendum to my formal comments here.
Major:
A schematic figure representing the different PE fragments and RITs that the authors discuss is needed in the interest of clarity.
-We added figure 1b to help the reader better understand the various exotoxin variants discussed in this review. (Page 4, Line 137)
The Perspective and Future Directions section seems lacking. Perhaps a summary of the immunogencity/half-life sections and some thoughts about bringing the anti-GPC3 RITs into the clinic would help to fill it out. It would also be helpful to emphasize and expand on what has changed since the last review.
-We restructured the Perspective and Future Direction section. We added summary paragraph to start this section that described the recent advances in HN3 immunotoxins. (Page 7, Line 283)
Minor:
Table 1: formatting is problematic. Please revise so that words are not broken across multiple lines.
-Table 1 was reformatted and condensed. This fixed the issues with multiple lines and broken words. (Page 3, Line 113)
Line 87: Start a new paragraph
-A new paragraph was started (Page 2, Line 84)
Lines 124-126: It is not clear to me how an antibody against heparan sulfate would be specific for GPC3. Wouldn’t it react with any HS? Can you add some details to clarify the specificity of HS20?
-We agree with the reviewer that this statement needed clarification. Due to the fact that HN20 does not binding directly to the core protein, it is likely that it will have ability to bind off-target proteins that share the same heparan sulfate modifications of GPC3. We added the following statement to the review, “These heparan sulfate modifications are not limited to GPC3, so HS20’s therapeutic potential needs to be validated due to its potential off-tumor toxicities.” (Page 4, Lines 124-125)
Line 150: Start a new paragraph
-A new paragraph was started (Page 5, Line 154)
Line 156: Start a new paragraph
-A new paragraph was started (Page 5, Line 160)
Line 173: Start a new paragraph
-A new paragraph was started (Page 5, Line 177)
Lines 196-197: It is not entirely true that the removal of domain II did not affect RIT function. It depends on several factors such as cell type and target antigen. More accurately, one could say that domain II was not essential to immunotoxin function or something similar.
-We altered our text to improve the statement’s clarity. This text was added to the review, ”The removal of domain II deleted a large number of protease sites, resulting in an immunotoxin that was less susceptible to lysosomal degradation. This study showed that domain II was not essential for immunotoxin function, although some cell types showed higher sensitivities to immunotoxins containing domain II.” (Page 5, Line 197-200)
Line 202: Start a new paragraph
-A new paragraph was started (Page 6, Line 206)
Line 223: Don’t change the nomenclature here by referring to “LR”. Keep previous PE24 nomenclature used elsewhere in this manuscript on lines 205/207/209/220/222/224/230/232.
-We agree with the reviewer that it will be easier for the reader if the same nomenclature is used. We will refer to domain III only as PE24, and the B cell deimmunized domain III as mPE24. It should be easier to recognize now with the addition of Figure 1B. (Page 4, Line 137)
Line 225: Start a new paragraph
-A new paragraph was started (Page 6, Line 230)
Lines 249 & 250: please specify the origins of the different albumin-binding domains to clarify the abbreviations.
-The origin of each domain was clarified by modifying the following statement. “We tested an albumin binding domain derived from the Streptococcal protein G (ABD) and a second one isolated from a llama VHH antibody (ALB1) to identify which would provide the greatest benefit for our immunotoxin.” (Page 7, Line 250-252)
Lines 250 & 251: please specify the model system in which the half-life study was conducted. Nude mice?
-The term “in nude mice” was added (Page 7, Line 256 and Line 257)
Lines 271-272: These statements belong in the immunogenicity section.
-We ended up deleting this statement from the manuscript. We did not want to mention the humanization of llama nanobodies, since these studies are not currently being conducted in our lab. Additionally, we don’t know if humanization is required for camelid nanobodies.
Lines 273-279: This paragraph belongs in the future directions section.
-This section was moved to the Future Directions section. (Page 8, Line 306)
Lines 281-282: This sentence does not fit with the remainder of the paragraph.
-This line was deleted with the reworking of the Perspective and Future Direction Section
Lines 282-288: This information seems out of context and would be more appropriate after a brief summary of the immunogenicity section.
-This section will now follow the summary of HN3 immunotoxins advancements. (Page 8, Line 299)
Citations: #39 & #40 appear to be identical. Please confirm that they are separate references.
-The citations were updated. (Page 11, Line 454-459)
Reviewer 2 Report
The article "Development of Glypican-3 Targeting Immunotoxins for the Treatment of Liver Cancer: An Update" is very interesting and very valuable review. I require some comments on the following questions
Q) about the statement that an increase PK exposure is necessary to increase activity; may this higher exposure be related to higher incidence of adverse effect ?
Q p3) about mild adverse effects. Some ITs clinical trials have been discontinued due to safety reasons. Please better delucidate this aspect.
Q table 1) Some studies are old, if reported add date or explain in the text the reason for study stop or discontinuing
Q p 15) About Pegylation. Pegylation is a widely used approach to increase PK of protein and nano therapeutics, nevertheless some evidences demonstrated that Peg linkage of immunogenic proteins can increase their immunogenicity. FDA highlighted the necessity of monitoring anti-Peg antibodies in clinical trials. Please comment.
Q) add a brief comment on the increasing approval of ADC in comparison of IT
Author Response
Comments and Suggestions for Authors Reviewer 2
The article "Development of Glypican-3 Targeting Immunotoxins for the Treatment of Liver Cancer: An Update" is very interesting and very valuable review. I require some comments on the following questions
- Q) about the statement that an increase PK exposure is necessary to increase activity; may this higher exposure be related to higher incidence of adverse effect ?
-We added the following statement to the text. “It is important to note that the increase in serum half-life also appears to be associated with increased off-target toxicities. The maximum tolerated doses we observed for the albumin binding immunotoxins was significant less than the original HN3-T20 immunotoxin.” (Page 7, Line 257-260)
Q p3) about mild adverse effects. Some ITs clinical trials have been discontinued due to safety reasons. Please better delucidate this aspect.
-We added a statement about the more severe side effects associated with PE-based immunotoxin treatment. This statement was added to the text, “In some PE-based clinical trials there have been adverse effects reported following treatment. Cardiac arrythmias, pneumonitis, hemolytic uremic syndrome, elevated liver enzymes and sepsis are some of the more severe side effects associated with immunotoxin treatment73-75. (Page 3, Line 96-99)
- Alewine, C.; Ahmad, M.; Peer, C. J.; Hu, Z. I.; Lee, M. J.; Yuno, A.; Kindrick, J. D.; Thomas, A.; Steinberg, S. M.; Trepel, J. B.; Figg, W. D.; Hassan, R.; Pastan, I., Phase I/II Study of the Mesothelin-targeted Immunotoxin LMB-100 with Nab-Paclitaxel for Patients with Advanced Pancreatic Adenocarcinoma. Clin Cancer Res 2020, 26 (4), 828-836.
- Mussai, F.; Campana, D.; Bhojwani, D.; Stetler-Stevenson, M.; Steinberg, S. M.; Wayne, A. S.; Pastan, I., Cytotoxicity of the anti-CD22 immunotoxin HA22 (CAT-8015) against paediatric acute lymphoblastic leukaemia. Br J Haematol 2010, 150 (3), 352-8.
- Foss, F. M.; Sjak-Shie, N.; Goy, A.; Jacobsen, E.; Advani, R.; Smith, M. R.; Komrokji, R.; Pendergrass, K.; Bolejack, V., A multicenter phase II trial to determine the safety and efficacy of combination therapy with denileukin diftitox and cyclophosphamide, doxorubicin, vincristine and prednisone in untreated peripheral T-cell lymphoma: the CONCEPT study. Leuk Lymphoma 2013, 54 (7), 1373-9.
Q table 1) Some studies are old, if reported add date or explain in the text the reason for study stop or discontinuing
-The table was condensed to highlight some of the more recent clinical trials. Mainly the table was included to highlight the various targets and toxins that are being tested in clinical trials. Those studies without published results has related publications cite. (Page 3, Lines 113)
Q p 15) About Pegylation. Pegylation is a widely used approach to increase PK of protein and nano therapeutics, nevertheless some evidences demonstrated that Peg linkage of immunogenic proteins can increase their immunogenicity. FDA highlighted the necessity of monitoring anti-Peg antibodies in clinical trials. Please comment.
-We agree with the reviewer that there are mixed reports about the immunogenic effect of PEGylation. We added the following statement to help clarify the text. “ However, increase IgM production following repeated doses of PEGylated liposomes results in accelerated blood clearance subsequent doses115. Additionally, anti-PEG antibodies can be detected in about 25% of healthy patients, most likely due to environmental exposure to PEG-containing compounds used in the cosmetics, pharmaceutical and food-processing industries116.” (Page 7, Line 276-280)
- Wang, X.; Ishida, T.; Kiwada, H., Anti-PEG IgM elicited by injection of liposomes is involved in the enhanced blood clearance of a subsequent dose of PEGylated liposomes. J Control Release 2007, 119 (2), 236-44.
- Garay, R. P.; El-Gewely, R.; Armstrong, J. K.; Garratty, G.; Richette, P., Antibodies against polyethylene glycol in healthy subjects and in patients treated with PEG-conjugated agents. Expert Opin Drug Deliv 2012, 9 (11), 1319-23.
- Q) add a brief comment on the increasing approval of ADC in comparison of IT
The following statement was added to highlight the recent approval of ADC therapeutics, “Antibody drug conjugates are another class of drugs that have seen a recent surge in FDA approvals44, 52.” (Page 2, Line 73-74).
- Birrer, M. J.; Moore, K. N.; Betella, I.; Bates, R. C., Antibody-Drug Conjugate-Based Therapeutics: State of the Science. J Natl Cancer Inst 2019, 111 (6), 538-549.
- Khongorzul, P.; Ling, C. J.; Khan, F. U.; Ihsan, A. U.; Zhang, J., Antibody-Drug Conjugates: A Comprehensive Review. Mol Cancer Res 2020, 18 (1), 3-19.